# Arterial Hypertension as a Risk Factor for Reduced Glomerular Filtration Rate after Living Kidney Donation

**DOI:** 10.3390/jcm9020338

**Published:** 2020-01-25

**Authors:** Julia Kerschbaum, Stefanie Bitter, Maria Weitlaner, Katrin Kienzl-Wagner, Hannes Neuwirt, Claudia Bösmüller, Gert Mayer, Stefan Schneeberger, Michael Rudnicki

**Affiliations:** 1Department of Internal Medicine IV, Nephrology and Hypertension, Medical University Innsbruck, Anichstraße 35, 6020 Innsbruck, Austria; Stefanie.Bitter@ordensklinikum.at (S.B.); m.weitlaner@tirol-kliniken.at (M.W.); hannes.neuwirt@i-med.ac.at (H.N.); gert.mayer@i-med.ac.at (G.M.); michael.rudnicki@i-med.ac.at (M.R.); 2Department of Visceral, Transplant, and Thoracic Surgery, Medical University Innsbruck, 6020 Innsbruck, Austria; katrin.kienzl@tirol-kliniken.at (K.K.-W.); claudia.boesmueller@tirol-kliniken.at (C.B.); stefan.schneeberger@tirol-kliniken.at (S.S.)

**Keywords:** arterial hypertension, living kidney donation, chronic kidney disease

## Abstract

Living kidney donation represents the optimal renal replacement therapy, but recent data suggest an increased long-term renal risk for the donor. Here, we evaluated the risk for reduced estimated glomerular filtration rate (eGFR), death, and major cardiovascular events such as nonfatal myocardial infarction or cerebrovascular event including TIA (transient ischemic attack) and stroke in 225 donors, who underwent pre-donation examinations and live donor nephrectomy between 1985 and 2014 at our center. The median follow-up time was 8.7 years (1.0–29.1). In multivariate analysis, age and arterial hypertension at baseline were significantly associated with a higher risk of adverse renal outcomes, such as (1) eGFR <60 mL/min/1.73 m^2^ (age per year: HR (hazard ratio) 1.05, 95% confidence interval (CI) 1.03–1.08, hypertension: HR 2.25, 95% CI 1.22–3.98), (2) eGFR <60 mL/min/1.73 m^2^ and a decrease of ≥40% from baseline (age: HR 1.08, 95% CI 1.03–1.13, hypertension: HR 4.22, 95% CI 1.72–10.36), and (3) eGFR <45 mL/min/1.73 m^2^ (age: HR 1.12, 95% CI 1.05–1.20, hypertension: HR 5.06, 95% CI 1.49–17.22). In addition, eGFR at time of donation (per mL/min/1.73 m^2^) was associated with a lower risk of (1) eGFR <60 mL/min/1.73 m^2^ (HR 0.98, 95% CI 0.97–1.00) and (2) eGFR <45 mL/min/1.73 m^2^ (HR 0.95, 95% CI 0.90–1.00). Age was the only significant predictor for death or major cardiovascular event (HR 1.08, 95% CI 1.01–1.16). In conclusion, arterial hypertension, lower eGFR, and age at the time of donation are strong predictors for adverse renal outcomes in living kidney donors.

## 1. Introduction

Living kidney donation is the optimal renal replacement therapy for patients with end-stage kidney disease in terms of life expectancy, renal function, and quality of life [1,2,3]. However, recent data suggest an increased long-term renal risk for the donor [4]. Two recent large cohort studies reported a higher risk of end-stage renal disease (ESRD) among donors compared with healthy nondonors [5,6]. Although ESRD has important implications on morbidity and mortality of the patient, it is a rare event, particularly in living kidney donors. Only limited data exist on more frequent intermediate outcomes, which are established risk factors for ESRD, such as decreased estimated glomerular filtration rate (eGFR). Ibrahim et al. [7] reported the risk of a reduced eGFR in 3956 kidney donors. After a mean follow-up time of 16.6 years, 36% reached an eGFR <60 mL/min/1.73 m^2^ and 2.8% had an eGFR <30 mL/min/1.73 m^2^ or had ESRD. In addition, 6.1% of the donors developed proteinuria. Risk factors for a decreased GFR were older age, a higher body mass index (BMI), and a higher systolic blood pressure. In the Swiss Organ Living-Donor Health Registry, occurrence of albuminuria increased from 4.8% to 10.4% over a follow-up period of 10 years. At last follow-up, the rate of microalbuminuria was significantly higher in donors who were hypertensive at the time of donation as compared to normotensive donors (16.6% vs. 6.0%, *p* = 0.03). Interestingly, urinary albumin excretion rate at the time of donation was not dependent on the presence of hypertension but was only associated with donor age [8].

The KDIGO (Kidney Disease: Improving Global Outcomes) Clinical Practice Guideline on the Evaluation and Care of Living Kidney Donors [9] summarizes that donor candidates with controlled hypertension (i.e., those having a systolic blood pressure less than 140 mmHg and diastolic blood pressure less than 90 mmHg using 1 or 2 antihypertensive agents and not having evidence of target organ damage) may be acceptable for donation. However, the decision to approve a donor candidate with hypertension has to be put into context with the transplant program’s threshold for long-term risk of end-stage renal disease and with the estimated life expectancy of the donor.

In this retrospective study, we identified risk factors for reduced eGFR after donation in 225 donors, who underwent living donor nephrectomy between 1985 and 2014 at our center. Furthermore, we also analyzed the risk of a combined endpoint of major cardiovascular events and death in this population.

## 2. Materials and Methods

### 2.1. Donors

Between January 1985 and March 2014, 262 persons living in Austria or South Tyrol (Italy) underwent pre-donation examinations and subsequently unilateral nephrectomy for living kidney donation at our center. Follow-up data were obtained in 234 of the donors (89.3%), the remaining 28 donors were lost to follow-up. According to the Austrian Central Register of Residence Registrations, the missing 28 donors were alive at the time of data collection. None of these donors had renal replacement therapy according to the Austrian dialysis and transplant registry, which covers nearly 100% of all dialysis patients in Austria. Data were collected retrospectively from the electronic patient charts. Serum creatinine value was obtained, and eGFR at baseline and at follow-up were calculated by the MDRD (modification of diet in renal disease) formula. In the earlier years of our study we analyzed creatinine clearance. Later, measured GFR (mGFR) was calculated based on Tc-99 m DTPA (diethylene-triamine-pentaacetate) measurements. Proteinuria and albuminuria were collected from 24-h urine measurements or from spot urine protein/albumin to creatinine ratios. Of the donors, 68.4% had no proteinuria or albuminuria at baseline, 3.0% had proteinuria or microalbuminuria, and in 28.6% no data could be obtained. As our study investigated a long time period of living kidney transplantation (1985–2014), criteria for the acceptance of living kidney donors have changed over time, and donors were accepted according to current definitions. In the early years of our program, there were 9 donors with an estimated GFR of below 60 mL/min/1.73 m^2^. All of these donors had a creatinine clearance or mGFR of >60 mL/min, and they were accepted for donation due to the lack of other comorbidities. To avoid biased results arising by inclusion of these donors, they were excluded from all further analyses. Hence, the number of donors included in this study was 225. Smoking was defined as active smoker or former smoker versus nonsmoker. Arterial hypertension was defined as prescription of antihypertensive medication (maximum of two antihypertensives) or having a blood pressure >130/80 mmHg in 24 h-ABPM (ambulatory blood pressure management) or two consecutive values >140/90 mmHg in office measurements.

### 2.2. Outcomes

Different renal outcomes after living kidney donation were evaluated, which were defined as (1) an eGFR <60 mL/min/1.73 m^2^, (2) an eGFR <60 mL/min/1.73 m^2^ and a decrease of eGFR ≥40% from baseline, and (3) an eGFR <45 mL/min/1.73 m^2^, respectively. Cardiovascular outcomes were defined as death or a major cardiovascular event (nonfatal myocardial infarction or cerebrovascular event including TIA (transient ischemic attack) and stroke).

### 2.3. Statistical Analysis

Nominal variables are shown as percent (%) and metric variables are shown as median (and minimum to maximum). Nominal variables were compared using the chi-square test, and metric variables were compared using the Mann–Whitney U test. Baseline parameters used in the univariate analysis were age, gender, body mass index (BMI), eGFR, arterial hypertension, smoking status, and relationship to the recipient. We also used Kaplan Meier plots and log rank test for testing univariate associations. All covariates showing univariate association with the outcome variable with a *p*-value less than 0.1 were entered into a multivariable Cox regression model. Results were expressed as hazard ratios (HRs) with 95% confidence intervals (95% CIs).

The Institutional Review Board of the Medical University Innsbruck approved this study (Ethic Committee approval code: AN2016-0145 364/4.10 on 20 September 2016).

## 3. Results

### 3.1. Baseline and Follow-Up Characteristics

Baseline characteristics of the 225 donors are summarized in Table 1. Median age of donors was 46 years (range 21–71) and 65.8% were female. A majority of donors (50.2%) had normal weight (BMI 20–<25 kg/m^2^), 40.9% had BMI values ≥25 kg/m^2^. Median eGFR was 86 mL/min/1.73 m^2^ at the time of donation (range 60–162). Approximately two-thirds of donors were related to the recipient (67.1%). Thirty-six percent of donors were smokers. Arterial hypertension was present in 15.6% of donors, and 10.2% had antihypertensive treatment (1–2 antihypertensive medications).

The median follow-up time was 8.7 years (range 1.0–29.1). Median eGFR at follow-up was 60 mL/min/1.73 m^2^ (range 18–122). No cases of end-stage renal disease occurred. At follow-up, 41.8% of donors had arterial hypertension. Death of any cause or a major cardiovascular event occurred in 4.4% of donors, and baseline eGFR did not differ significantly between donors with or without this endpoint (79 vs. 85 mL/min/1.73 m^2^, *p* = 0.677). Renal outcomes, i.e., (1) eGFR <60 mL/min/1.73 m^2^, (2) eGFR <60 mL/min/1.73 m^2^ and decrease of ≥40% from baseline, and (3) eGFR <45 mL/min/1.73 m^2^, occurred in 41.3%, 12.4%, and 7.6%, respectively. Proteinuria or albuminuria occurred in 3 donors after donation. However, we had no data on proteinuria/albuminuria for approximately 50% of the donors during follow-up, so we were not able to analyze risk factors for this indicator of incident chronic kidney disease.

### 3.2. Risk Factors for Adverse Renal Outcomes, Major Cardiovascular Events, and Death

In univariate analysis, the presence of arterial hypertension at baseline was significantly associated with adverse renal outcomes, i.e., (1) eGFR <60 mL/min/1.73 m^2^, (2) eGFR <60 mL/min/1.73 m^2^ and loss of ≥40% from baseline and, (3) eGFR <45 mL/min/1.73 m^2^ (Figure 1 and Figure 2, Appendix A).

In multivariate models, only age, eGFR, and the presence of arterial hypertension at baseline were significant predictors for adverse renal outcomes, i.e., (1) eGFR <60 mL/min/1.73 m^2^, (2) eGFR <60 mL/min/1.73 m^2^ and decrease of ≥40% from baseline, and (3) eGFR <45 mL/min/1.73 m^2^ (Table 2). Age per year was persistently associated with a higher risk for adverse renal outcomes (HR 1.05, 95% CI 1.02–1.07 for eGFR <60 mL/min/1.73 m^2^; HR 1.08, 95% CI 1.03–1.13 for <60 mL/min/1.73 m^2^ and loss of ≥40% from baseline; HR 1.11, 95% CI 1.04–1.19 for eGFR <45 mL/min/1.73 m^2^). Arterial hypertension showed a graded, significant association with adverse renal outcomes (HR 2.35, 95% CI 1.32–4.18 for eGFR <60 mL/min/1.73 m^2^; HR 3.95, 95% CI 1.60–9.73 for <60 mL/min/1.73 m^2^ and loss of ≥40% from baseline; HR 5.66, 95% CI 1.59–20.06 for eGFR <45 mL/min/1.73 m^2^). The eGFR (per mL/min/1.73 m^2^) was also significantly associated with (1) eGFR <60 mL/min/1.73 m^2^ and (2) eGFR <45 mL/min/1.73 m^2^ (HR 0.98, 95% CI 0.97–1.00 and HR 0.95, 95% CI 0.90–1.00). Addition of the number of classes of antihypertensive medications (none, one, or two medications) did not further improve the precision of the model (data not shown).

The only significant risk factor associated with death or a major cardiovascular event during follow-up was age (HR 1.08, 95% CI 1.01–1.16) (Table 2 and Appendix A).

### 3.3. Hypertension as a Risk Factor for Renal Outcomes

Donors with arterial hypertension at baseline were significantly older (median age: 55 years, range 30–68 years vs. 45, range 21–71, *p* < 0.001), had a higher BMI (26.6, range 17.4–34.2 vs. 24.2, range 17.4–38, *p* = 0.020), and were less frequently related to the recipient compared to donors without hypertension (48.6% vs. 70.5%, *p* = 0.011) (Table 3).

Median eGFR at follow-up was significantly lower in donors with hypertension (52 mL/min/1.73 m^2^, range 30–79 vs. 62 mL/min/1.73 m^2^, range 18–122), *p* < 0.001). Adverse renal outcomes occurred significantly more frequent in donors with arterial hypertension (76.9% vs. 44.8%, *p* = 0.002 for eGFR <60 mL/min/1.73 m^2^; 40.0% vs. 11.3%, *p* = 0.001 for eGFR <60 mL/min/1.73 m^2^ and loss of ≥40% from baseline; and 19.2% vs. 7.4%, *p* = 0.064 for eGFR <45 mL/min/1.73 m^2^).

## 4. Discussion

The findings of our study suggest that the presence of arterial hypertension at the time of living kidney donation, irrespective of the use of antihypertensive agents, represents a long-term risk factor for new-onset chronic kidney disease post-donation. During a median follow-up time of 8.7 years, 41.3% of the donors developed an eGFR <60 mL/min/1.73 m^2^. These numbers have to be interpreted with caution due to the retrospective study design, but they are in the range of the prevalence of 36% within 10 years post-donation reported by Ibrahim et al. for mostly Caucasian North American donors [7].

Reported risk factors for post-donation eGFR <60 mL/min/1.73 m^2^ include not only hypertension but also older age, increased body-mass index, and female gender in some but not all cohorts [7,10,11]. Only few papers on the proportion of donors with even lower post-donation eGFR have been published. In 2016, Ibrahim and colleagues showed that also the clinically more relevant endpoint of an eGFR <45 mL/min/1.73 m^2^ was associated with older age, a higher BMI, and a higher systolic blood pressure (BP) at donation [7]. Doshi et al. reported on the renal outcome in African American live kidney donors. After a mean follow-up of 6.8 ± 2.3 years, 6 of the 106 donors (6%) but none of the matched 235 healthy nondonors had an eGFR <45 mL/min/1.73 m^2^. However, the authors did not analyze any pre-donation risk-factors associated with this low eGFR [12].

The impact on health outcomes of living kidney donors with a post-donation eGFR <60 mL/min/1.73 m^2^ is still a subject of controversial discussions. On the one hand, the association of a reduced eGFR with an increased relative risk for various endpoints has been clearly established in large general population cohorts. It has been shown that eGFR <60 mL/min/1.73 m^2^ is associated with increased risk for all-cause and cardiovascular mortality, progressive chronic kidney disease (CKD), end-stage renal disease, and acute kidney injury even with an albumin-to-creatinine ratio (ACR) <30 mg/g [13]. It has been shown that living kidney donors have an 8- to 9-fold increased relative risk of end-stage kidney disease as compared to healthy controls, although the absolute long-term risk remains extremely low (0.5 events per 1000 person-years) [14]. On the other hand, the term “disease” may not be appropriate for donors who have no progressive deterioration of renal function, no other sign of kidney disease (such as albuminuria), and no or only few comorbidities [15]. To our knowledge only one paper has been published addressing the effect of post-donation eGFR on clinical outcomes. Ibrahim et al. analyzed data from 3956 mostly white kidney donors with a follow-up of 16.6 ± 11.9 years (the actual mean follow-up time differed between the various endpoints due to varying percentage of available measurements or self-reports) [7]. Post-donation eGFR <60 mL/min/1.73 m^2^ more than quadrupled the risk for death (HR 4.62, 95% CI, 3.70–5.77) and for the combined endpoint of an eGFR <30 mL/min/1.73 m^2^ or ESRD (HR 4.22, 95% CI, 2.65–6.71). Post-donation eGFR <45 mL/min/1.73 m^2^ was associated with an even higher relative risk of reaching an eGFR <30 mL/min/1.73 m^2^ or ESRD (HR 6.82, 95% CI, 4.19–11.11). Finally, an eGFR <30 mL/min/1.73 m^2^ after kidney donation was associated with a threefold increased risk of death (HR 2.99, 95% CI, 1.96–4.58). In summary, within 10 years after donation 36% of the living kidney donor cohort reported by Ibrahim had an eGFR <60 mL/min/1.73 m^2^, 11% had an eGFR <45 mL/min/1.73 m^2^, and 3% had an eGFR <30 mL/min/1.73 m^2^. Body mass index and systolic blood pressure at the time of donation were significantly associated with all of these adverse clinical outcomes. Furthermore, a post-donation decrease of eGFR increased the risk for death, an eGFR <30 mL/min/1.73 m^2^, or ESRD.

The kidney has a substantial renal functional reserve (RFR) capacity to maintain kidney function after loss of renal mass. The mechanisms behind RFR include adaptive hyperfiltration due to hyperperfusion (i.e., vasodilation and increase in renal plasma flow) and hypertrophy of the remaining glomeruli, but not glomerular hypertension [16]. Consequently, GFR is maintained after kidney donation but at the cost of a decrease in RFR. RFR can be estimated from the renal hemodynamic response after intravenous administration of either low-dose dopamine or an amino-acid solution, or after an oral protein load [17,18,19]. In living kidney donors it has been shown that older age and a BMI >30 kg/m^2^ were associated with a loss of RFR after donation [20]. Recently, it was also reported that overweight (i.e., BMI >25 kg/m^2^), young, female kidney donors have a significant decrease in RFR after donation [21]. No data exist on RFR in hypertensive living kidney donors, but Pecly et al. reported that RFR is significantly reduced in obese but not in lean hypertensive patients [22]. Interestingly, data from Zitta et al. suggest that RFR can be positively modified by antihypertensive treatment [23]. In summary, several factors, such as age and BM, reduce RFR and thus may influence long-term GFR after kidney donation. Our data raise the hypothesis that donors with hypertension at the time of donation may have a reduced RFR post donation, presumably due to a reduced or a loss of renal vasodilatory capacity. This impairs the renal compensatory response to adapt to a single-kidney state and results in a lower GFR during follow-up.

In our study, arterial hypertension at time of donation was not significantly associated with death or major cardiovascular events, which is in line with data by Mjoen et al. [24]. Although the latter study reported an increased all-cause and cardiovascular mortality risk in 1901 living kidney donors over a follow-up period of 15.1 years, systolic blood pressure at time of donation was also not associated with increased risk for these cardiovascular endpoints. We can speculate that the association of systolic blood pressure with increased cardiovascular risk cannot be easily detected in the relatively healthy populations of living kidney donors, particularly in small cohorts and with limited follow-up time.

The 2017 KDIGO Clinical Practice Guideline on the Evaluation and Care of Living Kidney Donors [9] (and also other guidelines for the evaluation of living kidney donors) recommends that donor candidates with controlled hypertension (defined as systolic blood pressure less than 140 mmHg and diastolic blood pressure less than 90 mmHg using 1 or 2 antihypertensive agents), who do not have evidence of target organ damage, may be acceptable for donation. However, the results of our analysis and data from other authors suggest that hypertension at the time of donation may be associated with a decreased post-donation eGFR independent of other risk factors. Therefore, we suggest to carefully assess potential living kidney donors with hypertension, in particular at a young age.

Limitations of our study are the retrospective design and, therefore, the number of losses to follow-up. A further limitation is the retrospectively calculated eGFR by MDRD formula given the fact that, in a proportion of donors, measured GFR differed from estimated GFR. Due to the long time span of almost 30 years, we were not able to find some baseline parameters in a sufficient proportion of donors such as albuminuria or proteinuria at the time of donation.

In our cohort, 31.3% of donors developed new-onset hypertension during follow-up, which is in range with the published data (approximately 40% after a mean follow-up of 16.6 years [7]). However, incidence of reduced eGFR and hypertension in donors with long follow-up times, for example, more than 15 years, have to be interpreted with caution. Donors who develop lower levels of eGFR and comorbidities, such as hypertension, are more likely to be followed in a specialized center, hence selection bias might be present.

The result that age at the time of donation was a strong predictor for reduced eGFR at follow-up can partly be explained by the fact that older age per se is a risk factor for chronic kidney disease. However, hypertension seemed to be an even stronger risk predictor especially in those donors who were under the age of 55 years at the time of donation (Appendix A).

The association of a higher pre-donation eGFR and the reduced risk for two of the renal endpoints (eGFR <60 mL/min/1.73 m^2^ and eGFR <45 mL/min/1.73 m^2^) has also been found by Ibrahim et al. [7] with similar hazard ratios to the ones found in our study (2–3% higher risk for the development of an increased eGFR per reduction of pre-donation eGFR of 1 mL/min/1.73 m^2^). Hence, it is reasonable to speculate that RFR is not only reduced in donors with arterial hypertension, but also in those with a lower eGFR at time of donation.

In conclusion, living kidney donation is the best treatment for ESRD but it remains a complex ethical and medical issue. We all expect that minimal risks for the donor should be outweighed by the psychological benefits of altruism and improved recipient health. Isolated medical anomalies such as hypertension may not be a contraindication to living kidney donation, but donors should be informed accordingly about potential risks and harms, and especially those donors with comorbidities should be followed carefully on a regular basis.

## Figures and Tables

**Figure 1 jcm-09-00338-f001:**
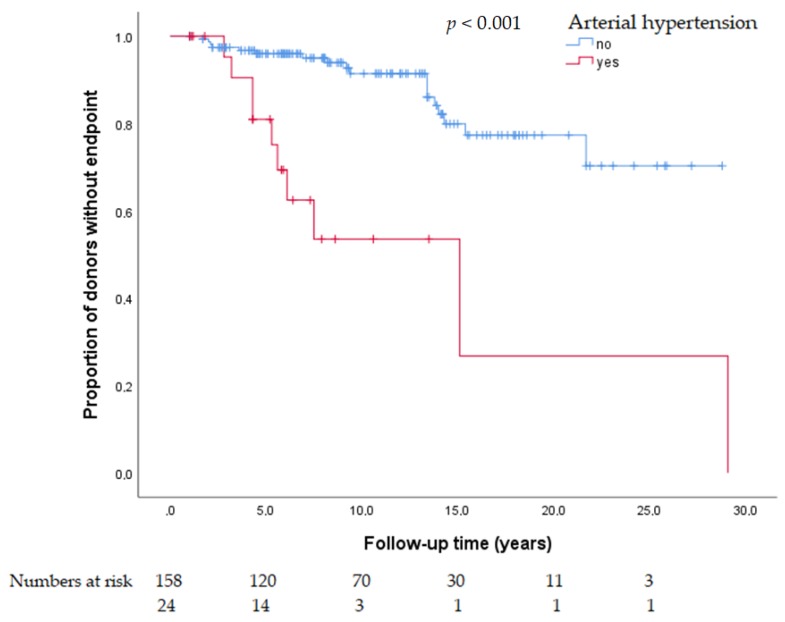
Kaplan Maier plot for arterial hypertension as a risk factor for estimated glomerular filtration rate (eGFR) <60 mL/min/1.73 m^2^ and decrease of ≥40% from baseline at follow-up.

**Figure 2 jcm-09-00338-f002:**
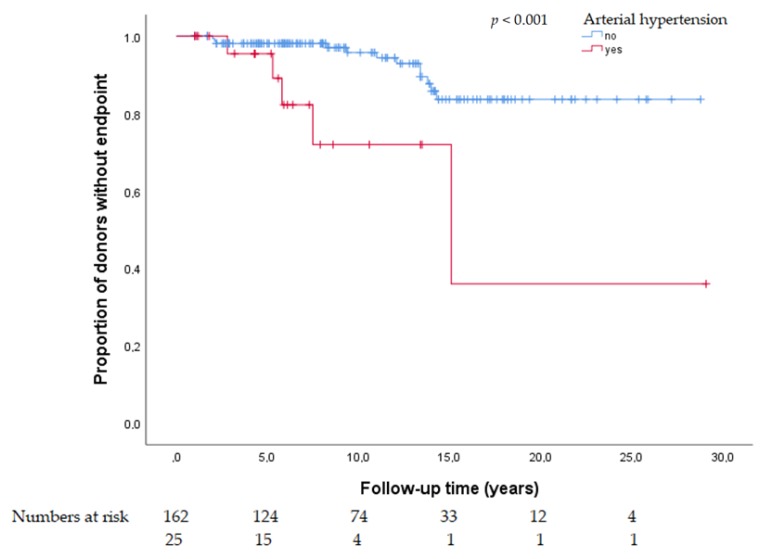
Kaplan Maier plot for arterial hypertension as a risk factor for eGFR <45 mL/min/1.73 m^2^ at follow-up.

**Table 1 jcm-09-00338-t001:** Baseline parameters of donors. Data are expressed as medians (minimum–maximum) and percentage. BMI means body mass index.

Baseline Parameters (*n* = 225)	
Age (years)	46 (21–71)
Female (%)	65.8
BMI (kg/m^2^)	24.5 (17.4–38.3)
eGFR (mL/min/1.73 m^2^)	86 (60–162)
Systolic blood pressure (mmHg)	120 (79–159)
Diastolic blood pressure (mmHg)	80 (44–105)
Related to recipient (%)	67.1
Smoking (%)	35.6
Arterial hypertension (%)	15.6
Antihypertensive therapy (%)	10.2
Follow-up time (years)	8.7 (1.0–29.1)

**Table 2 jcm-09-00338-t002:** Risk factors for adverse clinical outcomes (HR, hazard ratio; CI, confidence interval; eGFR, estimated glomerular filtration rate).

Clinical Outcome	Risk Factor	HR (95% CI)	*p*-Value
eGFR <60 mL/min/1.73 m^2^ (*n* = 93)	Age (per year)	1.05 (1.02–1.07)	<0.001
	Arterial hypertension	2.35 (1.32–4.18)	0.004
	eGFR (per mL/min/1.73 m^2^)	0.98 (0.97–1.00)	0.032
eGFR <60 mL/min/1.73 m^2^ and loss of ≥40% from baseline (*n* = 28)	Age (per year)	1.08 (1.03–1.13)	0.001
	Arterial hypertension	3.95 (1.60–9.73)	0.001
eGFR <45 mL/min/1.73 m^2^ (*n* = 17)	Age (per year)	1.11 (1.04–1.19)	0.002
	Arterial hypertensione GFR (per mL/min/1.73 m^2^)	5.65 (1.59–20.06) 0.95 (0.90–1.00)	0.007 0.040
Death or major cardiovascular event (*n* = 10)	Age (per year)	1.08 (1.01–1.16)	0.019

**Table 3 jcm-09-00338-t003:** Baseline and follow-up parameters for donors with and without arterial hypertension at baseline. Data are expressed as medians (minimum–maximum) and percentage (*n* = 225).

	Without Hypertension (*n* = 190)	With Hypertension (*n* = 35)	*p*-Value
*Baseline parameters*			
Age (years)	45 (21–71)	55 (30–68)	<0.001
Female (%)	66.3	62.9	0.692
BMI (kg/m^2^)	24.2 (17.4–38.3)	26.6 (17.4–34.2)	0.018
eGFR (mL/min/1.73 m^2^)	87 (60–162)	82 (64–120)	0.747
Systolic blood pressure (mmHg)	120 (79–155)	138 (118–159)	<0.001
Diastolic blood pressure (mmHg)	80 (44–100)	84 (70–105)	<0.001
Related to recipient (%)	70.5	48.6	0.011
Smoking (%)	35.8	34.3	0.864
Antihypertensive therapy (%)	0.0	65.7	<0.001
*Follow-up parameters*			
eGFR (mL/min/1.73 m^2^)	62 (18–122)	52 (30–79)	<0.001
Arterial hypertension (%)	31.3	100	<0.001
eGFR <60 mL/min/1.73 m^2^ (%)	44.8	76.9	0.002
eGFR <60 mL/min/1.73 m^2^ and loss of ≥40% from baseline (%)	11.3	40.0	0.001
eGFR <45 mL/min/1.73 m^2^ (%)	7.4	19.2	0.064

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
