# Peer review of "Arterial Hypertension as a Risk Factor for Reduced Glomerular Filtration Rate after Living Kidney Donation"

_jcm, 2020, doi:10.3390/jcm9020338_

Round 1

Reviewer 1 Report

Authors present a very good retrospective analysis of risk factors associated with few important endpoints after living kidney donation. Paper may not be the most original of all times, but yet deals with very important issue and is well prepared. I have only few comments that are detailed below.

Abstract

Line 14: major cardiovascular events such as??? In materials and methods: non-fatal myocardial infarction or cerebrovascular event including TIA and stroke. Could this be written also here in abstract?

Line 16: Data from 234 donors (89.3 %) were included in our analysis. Exclusion criteria?

Introduction

Clear, sufficient review of literature, clear research question

Materials and methods

Line 60: Accurate follow-up data were obtained in 234 of these donors (89.3 %). Please explain accurate follow-up data.

Line 69-70: Arterial hypertension was defined as prescription of antihypertensive medication. What was allowed? Only monotherapy or two medicines? What were the criteria of acceptance for donation? See comment at results section.

Results:

Lines 94-96: Donors with an eGFR <60ml/min/1.73m2 showed a mGFR (measured GFR) or 94 creatinine clearance of 69 to 106 ml/min, donors with an eGFR >120 ml/min/1.73m2 had a mGFR or 95 creatinine clearance of 91 to 168 ml/min. What were the donor exclusion criteria. How was the GFR measured, Cr-EDTA??

Line 140: Median eGFR at follow-up was significantly lower than in donors without hypertension; lower in patients with normotensic donors I presume.

Tables and figures are good and clear. Table 2: (to editor) could CI be positioned so that data remains in one line?

Discussion is thorough. Results of this study are well addressed, and limitations are noticed. Conclusion is valid.

Author Response

Thank you very much for your critical review and fruitful remarks! Please find enclosed our answer to your comments and questions:

1)    Line 14: non-fatal myocardial infarction or cerebrovascular event including TIA and stroke was added

2)     Line 16: This was clarified: . Follow-up data were obtained in 234 of the donors (89.3 %), the remaining 28 donors were lost to follow-up.

3)  Line 60: This was clarified according to the comment above, the term “accurate” was deleted.

4)  Lines 69-70: This was clarified: Arterial hypertension was defined as prescription of antihypertensive medication (maximum of two antihypertensives) or having a blood pressure >130/80 mmHg in 24 h-ABPM or two consecutive values >140/90 mmHg in office measurements.

The criterion for acceptance of donors with arterial hypertension was the presence of controlled hypertension with a maximum prescription of two antihypertensive medications.

5)     Lines 94-96: mGFR was measured by the Tc-99m DTPA method. This was added.

The centre followed the Consensus Statement of the Amsterdam Forum on the Care of the Live Kidney Donor (Transplantation 2004), donors were accepted when the mGFR was ≥ 80 ml/min. Before this time, two donors with a mGFR of 75 and 76 ml/min without comorbidities were accepted for donation.

6)     Line 140: This was corrected.

Reviewer 2 Report

The authors reported Arterial hypertension as a risk factor for reduced  glomerular filtration rate after living kidney donation. 

Minor criticism:

The only significant risk factor associated with death or a major cardiovascular event during follow-up was age. Arterial hypertension was associated with adverse renal outcome. I suggest to clarify with caution this important point

Author Response

Thank you very much for your critical review and fruitful remarks! Please find enclosed our answer to your comments and questions:

The following paragraph was added to the discussion section:

In our study, arterial hypertension at time of donation was not significantly associated with death or major cardiovascular events, which is in line with data by Mjoen et al [24]. Although this study reported an increased all-cause and cardiovascular mortality risk in 1901 living kidney donors over a follow-up period of 15.1 years, systolic blood pressure at time of donation was also not associated with increased risk for these cardiovascular endpoints. We can speculate that the association of systolic blood pressure with increased cardiovascular risk cannot be easily detected in the relatively healthy populations of living kidney donors, in particular in relatively small cohorts and with limited follow-up time.

Reference:

Mjoen, G.; Hallan, S.; Hartmann, A.; Foss, A.; Midtvedt, K.; Oyen, O.; Reisaeter, A.; Pfeffer, P.; Jenssen, T.; Leivestad, T., et al. Long-term risks for kidney donors. Kidney Int 2014, 86, 162-167, doi:10.1038/ki.2013.460.